# Gene Regulatory Network of ETS Domain Transcription Factors in Different Stages of Glioma

**DOI:** 10.3390/jpm11020138

**Published:** 2021-02-17

**Authors:** Yigit Koray Babal, Basak Kandemir, Isil Aksan Kurnaz

**Affiliations:** 1Institute of Biotechnology, Gebze Technical University, 41400 Gebze, Kocaeli, Turkey; ykbabal@gtu.edu.tr; 2Department of Molecular Biology and Genetics, Baskent University, 06790 Ankara, Turkey; bkandemir@baskent.edu.tr

**Keywords:** Ets, Elk-1, PEA3, Ets-1, glioma, biomarker

## Abstract

The ETS domain family of transcription factors is involved in a number of biological processes, and is commonly misregulated in various forms of cancer. Using microarray datasets from patients with different grades of glioma, we have analyzed the expression profiles of various *ETS* genes, and have identified *ETV1, ELK3, ETV4, ELF4,* and *ETV6* as novel biomarkers for the identification of different glioma grades. We have further analyzed the gene regulatory networks of ETS transcription factors and compared them to previous microarray studies, where Elk-1-VP16 or PEA3-VP16 were overexpressed in neuroblastoma cell lines, and we identify unique and common regulatory networks for these ETS proteins.

## 1. Introduction

ETS proteins are present in metazoan lineages [1] and play a role in diverse biological processes. Intriguingly, ETS proteins also exhibit extensive overlaps in their tissue expression profiles, with many members of this superfamily having ubiquitous expression [2]. Not surprisingly, members of this family also tend to exhibit overlapping and sometimes redundant DNA binding, as analyzed by genome-wide occupancy and other assays [3].

The ETS (E26 transformation specific) domain transcription factor superfamily includes 27 members in humans in 11 subfamilies [2,4]. Taking its name from the founding member of this superfamily, namely oncogenic v-ets [5], ETS domain transcription factors are typically defined by their DNA binding domain, called the ETS domain, which binds the consensus motif 5′-GGA(A/T)-3′, called the ets motif of the ETS binding site (EBS) [6]. Their DNA binding property, as well as transactivation function, is regulated by the MAPK signaling pathway [7,8].

In addition to their roles in normal growth and development, ETS proteins are commonly involved in cancer formation and progression through the regulation of cell proliferation, adhesion, migration, or vascularization, as well as regulation of epithelial–stromal interactions and epithelial–mesenchymal transition [9]. The expressions of several ETS family members, such as PEA3, ETS-1, and ETS-2, are upregulated in tumors, playing a role in different aspects of tumorigenesis, including tumor initiation, epithelial-mesenchymal transition, metastasis, and angiogenesis [4,10]. In some cases, ETS members are amplified and/or rearranged, such as c-ETS1 in acute myelomonocytic leukemia, or undergo chromosomal relocations that result in fusions, like in the case of chromosomal translocation of 5′ TMPRSS2 to the ETS genes, resulting in TMPRSS:ERG fusion proteins in nearly half of prostate cancers, or chromosomal translocation that yields EWS-FLI1 fusion in Ewing sarcoma. The transcriptional potency of ETS proteins is also often increased in various cancers as a result of changes in protein–protein interactions, post-translational modifications, and/or protein stabilization [4].

Ets-1 was found to be overexpressed in breast cancer, which was reported to be associated with poor prognosis [10]. Ets-1 is not only a critical regulator of invasion [10], but is also involved in regulating cancer energy metabolism in ovarian and breast cancer cell lines [11]. It has also been shown to play a role in telomere maintenance through the regulation of hTERT expression [12]. T417 phosphorylation of Elk-1, a member of the ternary complex factor (TCF) subfamily, was found to be associated with the differentiation grade of colonic adenocarcinomas [13]. Additionally, in high clinical stage prostate cancer, ELK-1, not TCF members ELK3 or ELK4, was found to be associated with disease recurrence [14]. In fact, ELK1 expression was reported to be higher than ELK3 in many cancer cell lines, including brain, skin, and myeloid tumors and sarcomas [15]. PKCα expression, parallel to cell migration and tumorigenicity, of hepatocellular carcinoma was increased by MZF/Elk-1 transcription factor complex [16]. The PEA3 subfamily of ETS domain transcription factors was also involved in a number of cancers, such as in lung tumors with MET amplification, and PEA3 subfamily members were found to play a role in migration and invasion [17]. In colorectal carcinoma, PEA3 was shown to promote invasiveness and metastatic potential [18]. In ovarian cancer, the loss of repressors of the PEA3 subfamily was shown to cause overaccumulation of ETV4 and ETV5 [19].

Malignant gliomas are the most common and lethal primary tumors of the brain. Grading of diffuse gliomas is based largely on the mitotic activity and vascular proliferation states, and molecular markers are also used as diagnostic entities. Still, further molecular information will be important in a more detailed description and categorization of central nervous system (CNS) tumors, in particular for reliable and reproducible classification of grade II and grade III diffuse gliomas [20,21]. Glioblastoma multiforme (GBM), WHO grade IV, is the most aggressive and lethal among all gliomas. High-grade gliomas are composed of a highly proliferative tumor core, with highly invasive cells surrounding them [22].

ETV2, an early regulator of vascular development, was found to be overexpressed in high-grade gliomas, and was reported to play a critical role in endothelial transdifferentiation of CD133+ GBM stem cells, which is thought to render them resistant to anti-angiogenic therapy [23]. Another ETS-related gene, ERG, was found to be a novel and highly specific marker for endothelial cells within CNS tumors, a feature that can be used in studying the vascularization of gliomas [24]. A transposon-based study of gliomagenesis identified friend leukemia integration 1 transcription factor (Fli1), among other genes, to be expressed in gliomas, although Fli1 expression is limited to a subset of glioma cells [25], and ETS protein PU.1, known for its critical role in hematopoietic development, was also reported to be highly expressed in glioma patients, indicating its role in the progression of glioma [26]. In addition to their role in tumor vascularization, ETS proteins can also regulate other aspects of tumorigenesis. In a network analysis based on complexity, as measured by betweenness, Etv5 was identified as a regulator in low-grade optic gliomas in Nf1 mutant mice, and experiments validated the increased expression of both Etv5 and its target genes in optic gliomas [27].

Gliomas can be broadly classified as diffuse and non-diffuse (circumscribed) gliomas. Diffuse gliomas, namely oligodendrogliomas and astrocytomas, exhibit similarities to glial precursors, and are identified and categorized based on the WHO classification of CNS tumors [28]. Due to their rather heterogeneous nature, the reproducibility in diagnosis of low-grade (WHO grade II/III) diffuse gliomas can be a challenge. Several molecular markers, such as isocitrate dehydrogenase (IDH) mutations or telomerase reverse transcriptase (TERT) promoter mutations, which create ETS binding sites [12], are used to assist in differential diagnosis [20]. Previously, ETS gene status in clinical prostate tumor samples has been determined, and ERG+ and ETV1/4/5+ cases were found to be associated with worse prognosis, indicating that ETS status may act as a prognostic biomarker and be used in combination with other existing molecular determinants [29]. In this study, we have analyzed microarray data from patients with different grades of glioma for relative expression of ETS genes, and identified different ETS genes that are upregulated at different glioma grades. We show that, while ETV1 is expressed at high levels in grade 2 glioma, its expression gradually decreases with glioma stage, and on the other hand, ELK3 and ETV4 expressions are increased with progression of the glioma stage. Furthermore, both ELF4 and ETV6 expressions are downregulated at grade 2 glioma, but upregulated at increasing levels in grades 3 and 4, indicating that these genes can also be utilized as additional molecular determinants to distinguish glioma grades. We further compare these data to microarray results from Elk-1-VP16 or PEA3-VP16 overexpression SH-SY5Y cells in order to narrow down transcriptional regulons, and identify common and unique transcriptional regulatory networks for these ETS proteins. 

## 2. Materials and Methods

### 2.1. Data Collection

Microarray datasets related to glioma were searched from the Gene Expression Omnibus data repository [30], and GSE4290 datasets, including expression data of brain tissue from glioma patients, were selected in this study [31]. The dataset contains the brain tissue of three glioma grades (grade 2–4) from glioma patients and epilepsy patients as a non-tumor control by obtaining brain tissue from surgery patients from Henry Ford Hospital. Patient classification and tissue preparation for microarray were described in [30]. A preprocessed expression matrix was imported into an R programming interface by using R package GEOquery from the Bioconductor project [32]. The methodological flow chart of the study is shown in Figure 1.

### 2.2. Data Processing and Differentially Gene Expression Analysis

The expression matrix of the study was Log2 normalized, used in principal component analysis (PCA) to investigate the relationship between samples. Outlier samples were determined by using the PCA plot and eliminated from the expression matrix before the differential gene expression analysis. Differential gene expression analysis was performed with the R package limma from Bioconductor with the contrast of each glioma grade versus non-tumor samples [33]. Differentially expressed genes (DEGs) were determined with a Benjamini & Hochberg corrected *p*-value < 0.05 significance level and absolute Log2 fold change > 0.6. DEGs were visualized with a volcano plot by using R package EnhancedVolcano from Bioconductor [34]. Additionally, the intersection of DEGs was performed between limma results, Elk-1-VP16, and the PEA3-VP16 overexpression microarray, which were previously published from our laboratory for use in further analysis.

### 2.3. Gene Regulatory Network Construction

Transcriptional gene regulatory network (GNR) mediated by the ETS transcription factor family was constructed by using the R package RTN from Bioconductor [35,36,37]. The expression matrix and ETS transcription factor list from differentially expressed genes were used as an input for the transcriptional network inference (TNI) algorithm with a Benjamini & Hochberg corrected *p*-value < 0.05 significance level and 1000 permutation number parameters. Regulon activity scores were calculated from TNI by using a two-tailed gene set enrichment analysis (GSEA) algorithm built-in RTN package and visualized as a heatmap. To construct an edge-weighted gene regulatory network mediated with ETS members (Regulons), transcriptional regulatory networks from TNI and differentially expressed genes were integrated by a transcriptional regulatory analysis (TNA) algorithm with two-tailed GSEA. A gene regulatory network of ETS members was constructed by using significant network interaction from TNA (*p*-value < 0.05). Additionally, the constructed network was filtered with the intersection of DEGs and previous microarray studies (Elk-1-VP16 and PEA3-VP16 overexpression studies). The final GNR was visualized by using Cytoscape software [38].

### 2.4. Functional Enrichment Analysis

Gene ontology (GO) and KEGG pathway enrichment analysis were performed by using R package clusterProfiler from Bioconductor to analyze the functional profile of gene clusters from differentially expressed genes (up- and downregulated genes of individual glioma grade) and transcription factor regulon clusters (positively and negatively regulated targets for each ETS regulon) [39]. The enriched GO term and KEGG pathways were determined by using a Benjamini & Hochberg corrected *p*-value < 0.05 significance level and context manner term filtration.

## 3. Results

### 3.1. Identification of Differentially Expressed Genes in Gliomas

Before analyzing the expression of genes within the ETS superfamily, we have obtained and analyzed microarray datasets corresponding to different stages of glioma from patients (grades 2–4), using epilepsy patients as the non-tumor control [31].

Analysis of the transcriptome profiles of these glioma samples yielded a set of differentially expressed genes (DEGs), and the performance of the differential expression levels in discriminating the tumor cells from non-tumor cells was verified through principal component analysis (Figure 2). According to the PCA analysis, it was found that, while non-tumor and glioma samples were readily separated, discrimination of glioma stages was less pronounced; grade 2 and grade 4 gliomas were relatively separate, however, grade 3 tumor samples were not clearly separated from the other tumor groups (Figure 2A). Therefore, any outlier data were eliminated for differential gene expression analysis. According to differential gene expression analysis, 8402 of 19,225 genes were found to be significantly changed (with adjusted *p*-value < 0.05 and absolute log2 fold change > 0.6 thresholds) in the tumor samples (grades 2–4) (Appendix A). Up- and downregulated genes in all three grades of gliomas were obtained by differential gene expression analysis, and the volcano plot was used to visualize the DEGs, which shows that the distribution of differentially expressed genes was compatible (Figure 2B–D). 

We have next asked whether ETS genes were among the DEGs, and, if so, how their expressions were affected in different glioma stages compared to the non-tumor control (Figure 2E). To that end, we have focused our studies to ETS subfamilies that have been previously reported to be involved in gliomas, namely the ETS subfamily [40,41,42], TCF subfamily [43,44,45], ELF subfamily [46,47], PEA3 subfamily [27,48], and TEL subfamily [49,50]. The results showed that the expressions of ETS2 and ELK-1 were downregulated at all grades, while the expressions of ELK4, ELF1, ELK3, ETS1, ETV4, and ETV1 were upregulated at all grades; intriguingly, ETV6 and ELF4 expressions were downregulated at grade 2, but upregulated at grades 3 and 4. The expression of ELK3, ETV4, and to some extent ELK4 was found to increase gradually with glioma grade, while ETV1 expression was highest in grade 2 glioma, and progressively decreased with glioma stage (Figure 2E). The only member of the SPI subfamily of ETS transcription factors represented was PU.1, which was previously shown to be involved in glioma progression, and its levels were indeed found to be increased with glioma grades (Figure 2E; [26]).

Previous microarray studies where either constitutively active Elk-1-VP16 or PEA3-VP16 was overexpressed in SH-SY5Y neuroblastoma cells have identified a number of transcriptional targets for these ETS proteins. In order to narrow down our search and focus on gene regulatory networks of ETS proteins, we have identified overlapping genes by comparing DEGs from glioma grades 2–4 with the microarray results from Elk-1-VP16 and PEA3-VP16 overexpressing cells; 2637 genes were found to be at the intersections of these two experiments, with 63 genes commonly regulated in both glioma tumor samples (DEGs), as well as cell line studies (Elk-1-VP16 and PEA3-VP16) (Figure 2F).

To clarify the functional profiles of the identified DEGs in glioma grades, enrichment analysis was performed, and significant GO and KEGG annotations were obtained (Figure 3). For the GO enrichment analysis of biological processes, initially, up- and downregulated genes of the different glioma grades were analyzed. While the upregulated gene clusters of grades were observed with cell cycle related phenotype in all glioma grades, the downregulated gene clusters of grades showed neuronal phenotype, such as synaptic transmission, as would be expected (Figure 3A, Appendix A). In the KEGG enrichment analysis, while the upregulated genes of glioma grades were enriched in p53, TGF-β, and Notch signaling pathways, prominent downregulated gene clusters fell into synaptic function related pathways, such as glutamatergic, GABAergic, serotonergic, and dopaminergic pathways (Figure 3B, Appendix A). For instance, the TGF-β signaling pathway was found to be altered through glioma progression, as observed by an increase in the level of BMP molecules, including BMP3 and BMP4, as well as their targets, such as Smad1/5/8 and Id. Additionally, the expression of cMyc and p15 associated with cell cycle were significantly increased. On the other hand, Notch signaling pathway genes, such as Delta, Notch, and Fringe, were observed to be upregulated in glioma. The MAPK signaling pathway was found to be enriched in downregulated clusters, with the expression of genes, such as Ras, MEK1, ERK, JNK, and Elk-1, being downregulated. All of these functional enrichment analysis results confirmed that, in all of the glioma grades, cells had downregulated pathways directly related with neuronal function, but upregulated signaling pathways related to cell proliferation and survival. It is interesting to note that the grade 4 glioma samples did not exhibit significant TGF-β pathway upregulation, but PI3K pathway upregulation instead (Figure 3B).

### 3.2. Transcriptional Gene Regulatory Network Construction

In order to investigate ETS transcriptional regulation networks specific for each glioma grade, we have integrated DEGs obtained from the analysis of glioma grades with the normalized expression matrix, where significantly changed ETS members are referred as regulons. The initial network obtained using the transcriptional network inference (TNI) algorithm contained 10 ETS member regulons, 11,762 target genes, and 23,181 total interactions (Figure 4A). We have focused on the expression changes of ETS members in different grades of glioma, and regulon activity scores were calculated from the initial network. The analysis of regulon activity scores showed that, while ELK-1 and ETS2 showed high regulon activity in the non-tumor condition (magenta, Figure 4A), ETV1 showed a high activity score on mainly grade 2 glioma (green, Figure 4A). The other ETS proteins showed high regulon activity in the grade 4 glioma cluster, however, regulon activity of grade 3 glioma was dispersed between grade 2 and grade 4 (Figure 4A). After including DEGs into the initial network using the transcriptional network analysis (TNA) algorithm, a focused network of glioma grades was constructed. According to TNA algorithm, nine significant ETS regulons were found to be enriched with different numbers of target genes. ETV1, which expressed the least significance among the significant ETS regulons, was marked in blue (Figure 4B).

To determine the direction of regulation between each regulon and its targets, two-tailed GSEA was performed, and positively and negatively regulated co-expression patterns in target gene distribution were constructed for individual ETS regulons (Figure 5). In this analysis, genes were ranked for their fold changes in the x-axis, and enrichment scores were given in the y-axis; the peak of each plot is the enrichment score for the gene indicated (dotted lines), while the colored bar shows the positively and negatively correlated genes. These results suggest that enriched ETS regulons have both unique and common gene targets in gliomas, as indicated by a clear separation of negatively and positively correlated targets in regulons such as ETS2 and ELK1 (unique), and overlapping negative and positive regulons, such as those of ELF4 and ELK3 (common). 

The network from the TNA algorithm was filtered by DEGs from Elk-1-VP16 and PEA3-VP16 overexpression microarray results to create a much more unique regulatory network of ETS members. As a result of filtering, a final regulatory network was constructed with 3366 target genes and 6610 interactions with ETS regulons, and the gene regulatory network was visualized with Cytoscape (Figure 6). This network representation shows fold changes of DEGs, as well as their interaction with the ETS regulons, showing the common targets to be clustered in the middle (Figure 6). 

Focusing on the functional investigation of the gene regulatory network, GO and KEGG enrichment analysis was performed with positively and negatively regulated targets of ETS regulons on the regulatory network (positive regulation indicates similar coexpression patterns, i.e., when ETS protein is downregulated, its targets are also downregulated, and vice versa). It was observed that positive and negative cluster targets of ETS regulons were enriched in biological processes, such as cell–cell adhesion, synapse formation, and protein localization, some of which are common across members, while some are unique for one or few family member(s) (Figure 7A,B, Appendix A). ELF1 and ELF4 regulons appear to belong to similar biological processes; ELK1 and ETS2 also were found to have targets within the same biological pathways (Figure 7A). The ETV1 regulon appears to have a distinct set of positively regulated targets, while ELK3, ELK4, ETS1, and, to some extent, ETV4 appear to regulate similar biological processes (Figure 7A). This classification is not conserved for negatively regulated targets, however; here, ELF1, ELK3, ELK4, ETS1, and ETV4 appear to regulate similar biological processes, while the ELF4 regulon and ETS2 regulon each are comprised of distinct targets (Figure 7B). It is important to note that, in positively regulated targets of the ETV1 regulon, nucleosome and chromatin disassembly related processes were prominent, while no significant negatively regulated targets were identified for the ETV1 regulon. The ELK3 regulon included positively regulated targets in ECM organization, protein maturation, and processing pathways, and negatively regulated targets in synaptic vesicle signaling, synaptic transmission, and synaptic plasticity pathways.

Similar comparative analysis using KEGG pathway enrichment showed that, unlike the GO biological processes analyzed above, distinct signaling pathways were regulated by each ETS regulon, while a positively regulated cluster of the ETV4 regulon was enriched for the MAPK and PI3K-Akt signaling pathways, and a negatively regulated cluster of this protein was enriched for endocytosis and the synaptic vesicle cycle (Figure 8A,B, Appendix A); the positively regulated cluster of ELK1 was enriched for cholinergic and dopaminergic synapses, as well as calcium signaling, while its negatively regulated cluster was enriched Hippo signaling, signaling of pluripotent stem cells, and cell cycle (Figure 8A,B). Interestingly, endocytosis and the synaptic vesicle cycle were common signaling pathways in almost all ETS regulons, except for ELK1 (Figure 8B).

## 4. Discussion

The five stages of gliomagenesis are the initial growth stage, oncogene-dependent senescence stage, growth stage, replicative senescence stage, and, finally, the immortalization stage [28]. Disease stage classification and identification of stage-dependent or grade-dependent biomarkers is important in the accurate diagnosis of gliomas. 

Graph complexity analysis in low-grade glioma has shown Etv5 and its network expression to be critical features of the neoplastic state [27]. Unfortunately, ETV5 of the PEA3 subfamily does not appear to be significantly altered in tumor vs. non-tumor samples in the microarray datasets used in this study (data not shown). However, we have identified another PEA3 subfamily member, ETV1, to be expressed at high levels in low-grade glioma and decrease in expression in higher grades (Figure 2). ELK-1 protein is known to be a critical partner for the androgen receptor (AR) in prostate cancer, and its expression was found to be associated with a higher clinical stage and prognostic marker of disease recurrence in prostate cancer [14]. No such distinction was apparent in our study on glioma grades 2–4. However, we have identified ELF4 and ETV6 to be downregulated in grade 2 gliomas, and upregulated in increasing amounts in grades 3 and 4 (Figure 2). It should be noted, however, that the ETS expression profile is also different in epilepsy; ELF1, ELK1, ELK4, ETS1, ETS2, and ETV1 are expressed at higher levels than ELF4, ELK3, and ETV4, and there is also variability in expression among different types of epilepsy (Appendix A). However, since the type of epilepsy used in the datasets analyzed in this study were not known, it was not possible to normalize for ETS gene expression (Appendix A) [51].

ETS proteins focused on in this study (namely, class I subfamilies ETC, TCF, and PEA3 and class II subfamilies ELF and TEL) exhibit little tissue specificity, and, in fact, many family members are ubiquitously expressed [2,15]. It is therefore not surprising that gene regulatory network analysis of ETS transcription factors exhibits extensive overlap of targets, confirming that functional redundancy exists at least to a certain extent (Figure 6). However when positively and negatively regulated targets of ETS regulons were analyzed, negatively regulated targets were found to extensively overlap (mostly related to synaptic vesicle trafficking and synaptic transmission, Figure 7B), while positively regulated targets appeared to be selective for groups of ETS family members, with ELF1 and ELF4 comprising one class (targets in synapse pruning, immune function, and cell to cell adhesion related biological processes), ELK1 and ETS2 forming another class (targets in synapse function and synaptic vesicle trafficking related processes), and ELK3, ELK4, ETS1, and ETV4 forming a third class (targets in extracellular matrix related processes, as well as protein processing and localization, ER stress, and cell cycle related processes); ETV1 appeared to be a class by itself (targets in nucleosome and chromatin disassembly) (Figure 7A). These regulon classes were not directly related to ETS subfamily assignments. 

Although more physiologically relevant non-tumor controls are required to fine-tune the results in the future, this is a proof of concept study that shows that expression levels of ETS genes can be used as diagnostic markers for glioma grade identification, in addition to already existing molecular markers. In addition, the gene regulatory network analysis for ETS regulons can be used to identify target gene clusters in positively and negatively regulated pathways and processes, which can help in understanding the molecular mechanisms of transcriptional redundancy among family members. We propose that such network analysis can also be extended to differentiate stages of tumorigenesis in other types of tumors, as well as to developmental stages of various tissues.

## Figures and Tables

**Figure 1 jpm-11-00138-f001:**
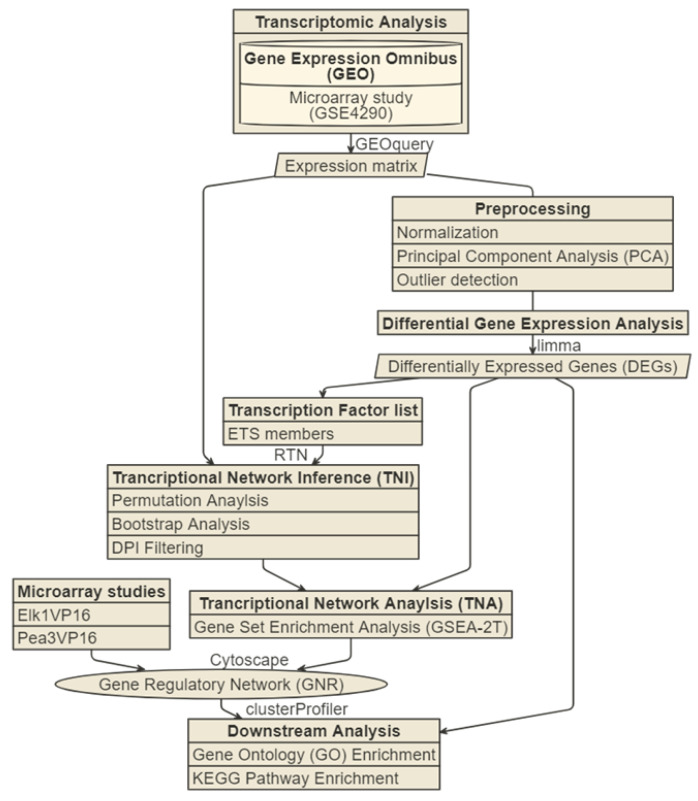
Methodology of transcriptomic profiling and gene regulatory network inference algorithm.

**Figure 2 jpm-11-00138-f002:**
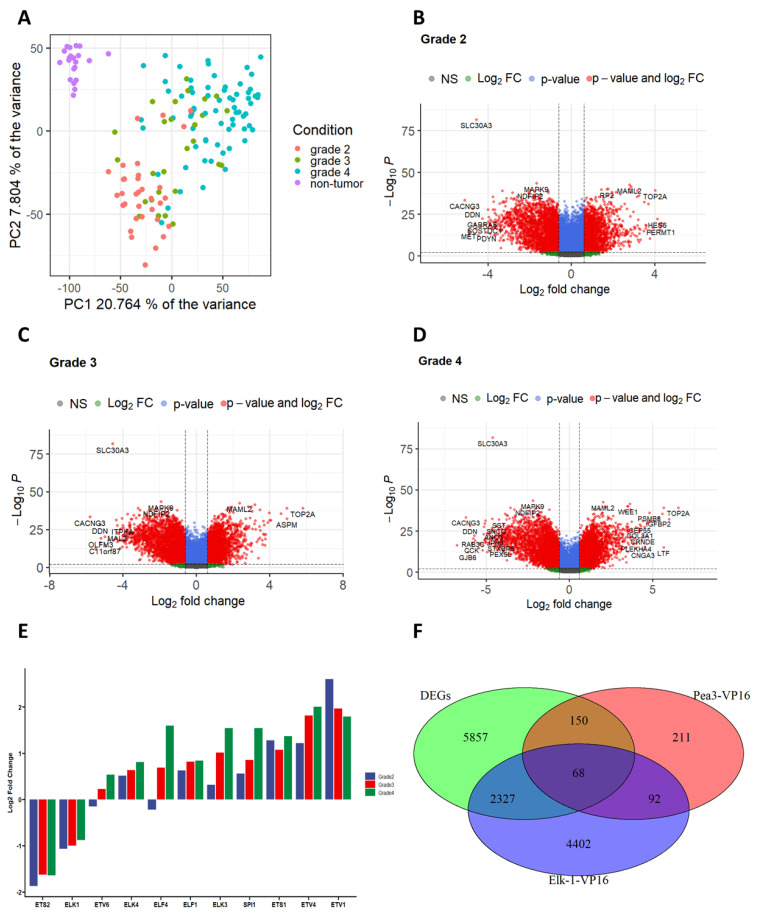
(**A**). Differential gene expression analysis results. Principal component analysis (PCA) plot of the normalized expression matrix. Each point represents individual samples. (**B**–**D**). Corresponding differentially expressed genes (DEGs) were obtained from comparisons of non-tumor vs. individual glioma grades by using the limma package with a 0.6 absolute log2 fold change and adjusted *p*-value < 0.01 with FDR cutoff, which is indicated with red points. (**E**). Relative fold change of significantly changed ETS members for glioma grade compared with non-tumor samples by differential gene expression analysis. (**F**). Intersection of DEGs between limma results, Elk1VP16 and PEA3VP16, which are our previously published micro arrays, were represented as venn diagrams.

**Figure 3 jpm-11-00138-f003:**
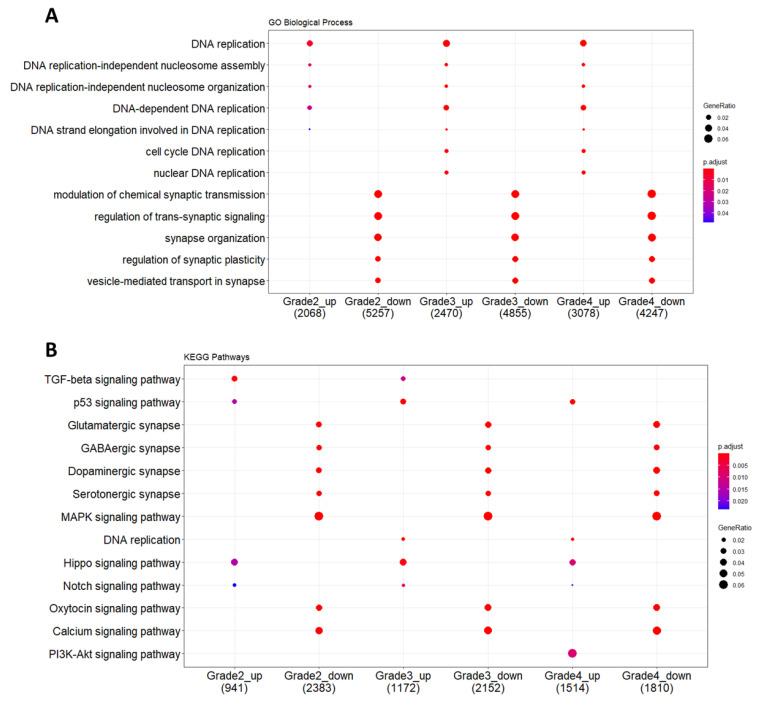
GO and KEGG enrichment analysis of differentially expressed genes for glioma grades. Each glioma grade was clustered as up- and downregulated gene clusters. (**A**). GO and (**B**). KEGG analysis were performed by clusterProfiler with an adjusted *p*-value < 0.05. The gene ratio indicates the number of genes enriched with a corresponding GO or KEGG term among the total gene number introduced into the enrichme.nt analysis.

**Figure 4 jpm-11-00138-f004:**
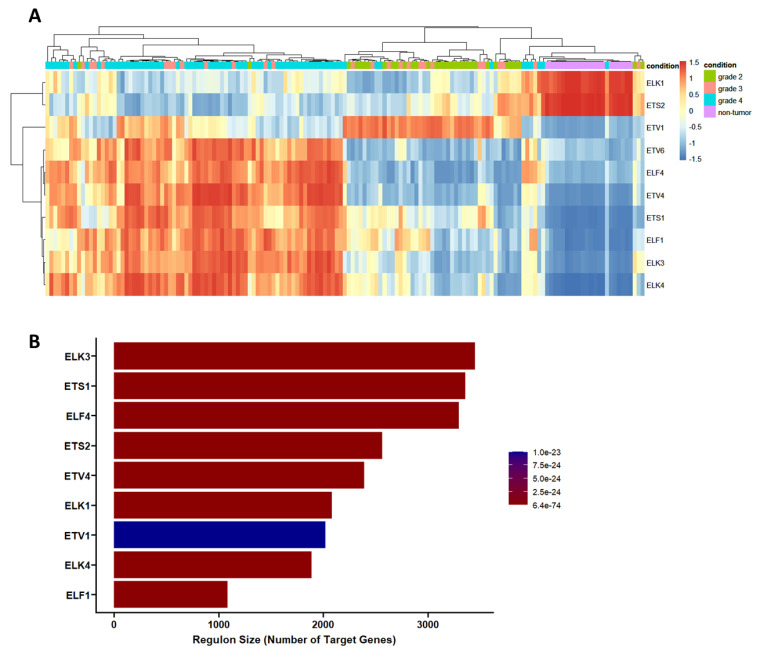
Regulon activity and size of the ETS-mediated gene regulatory network. (**A**). The correlation distance heatmap of regulon activity for non-tumor and glioma grades. (**B**). Regulon size of the individual transcription factor in the gene regulatory network resulting from the transcriptional network analysis (TNA).

**Figure 5 jpm-11-00138-f005:**
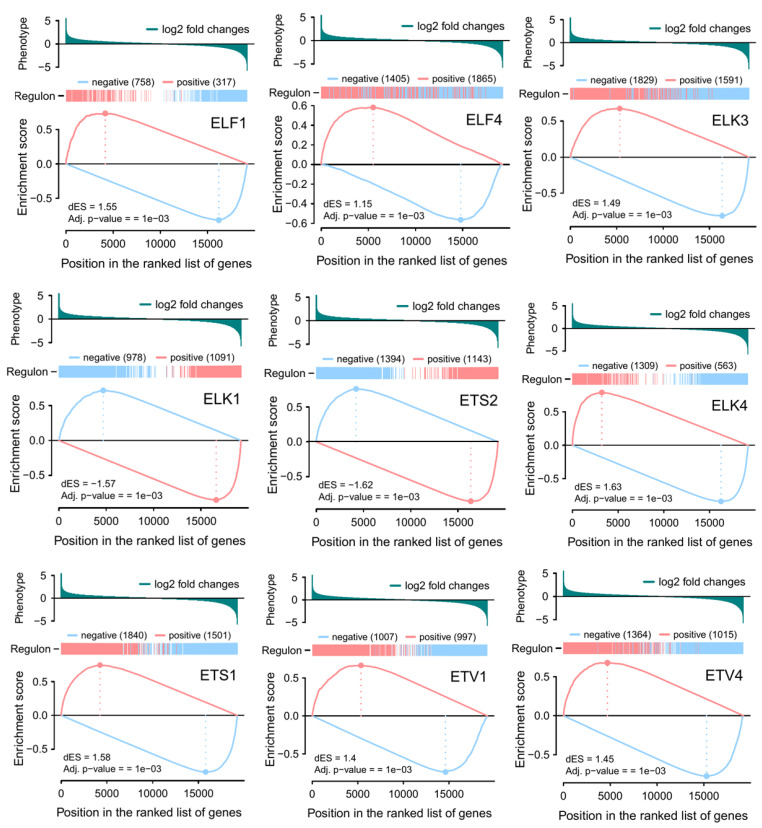
Two-tailed GSEA analysis associated positively and negatively regulated targets of individual regulons. Target genes are ranked by gene expression analysis, and scored by enrichment analysis that indicates the edge weight of the gene regulatory network.

**Figure 6 jpm-11-00138-f006:**
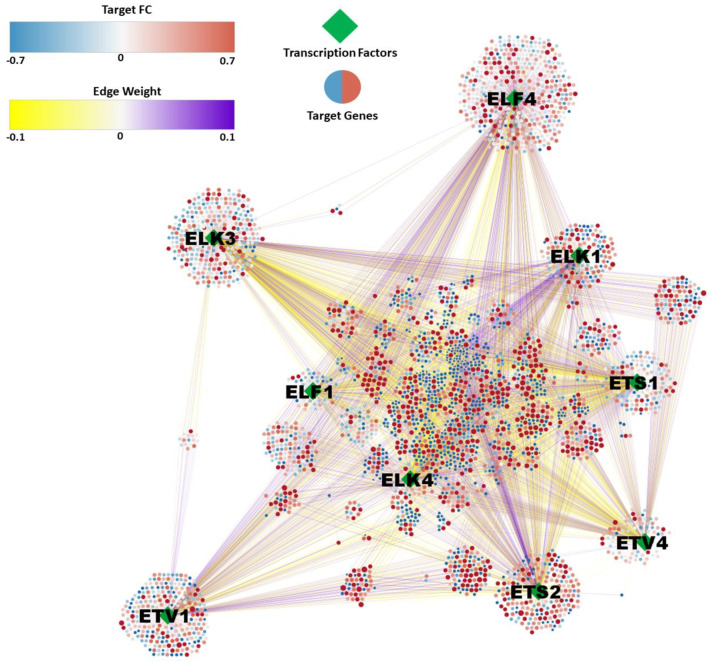
Gene regulatory network of glioma grades under the regulation of ETS transcription factors. Diamond nodes represent ETS members as a regulator, and circle nodes correspond to the target genes. The color of the circle indicates the mean fold change of glioma grades compared to non-tumor samples, resulting from differential gene expression analysis. Edge colors show the enrichment score of each target gene with corresponding regulators resulted from GSEA analysis.

**Figure 7 jpm-11-00138-f007:**
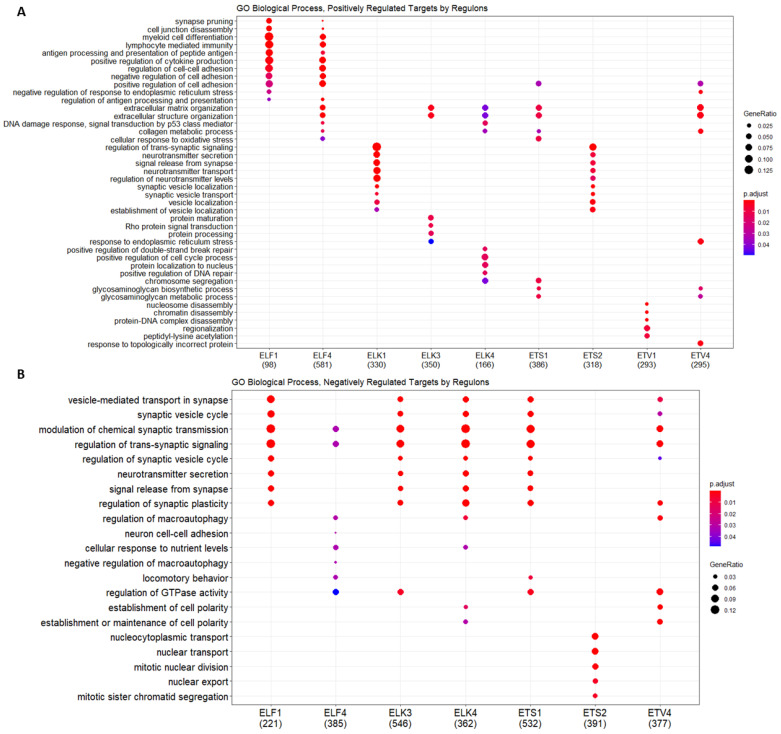
GO enrichment analysis of individual regulons and their (**A**). positively and (**B**). negatively regulated targets. GO analysis was performed by clusterProfiler with an adjusted *p*-value < 0.05. The gene ratio indicates the number of genes enriched with corresponding GO terms among the total gene number introduced into the enrichment analysis.

**Figure 8 jpm-11-00138-f008:**
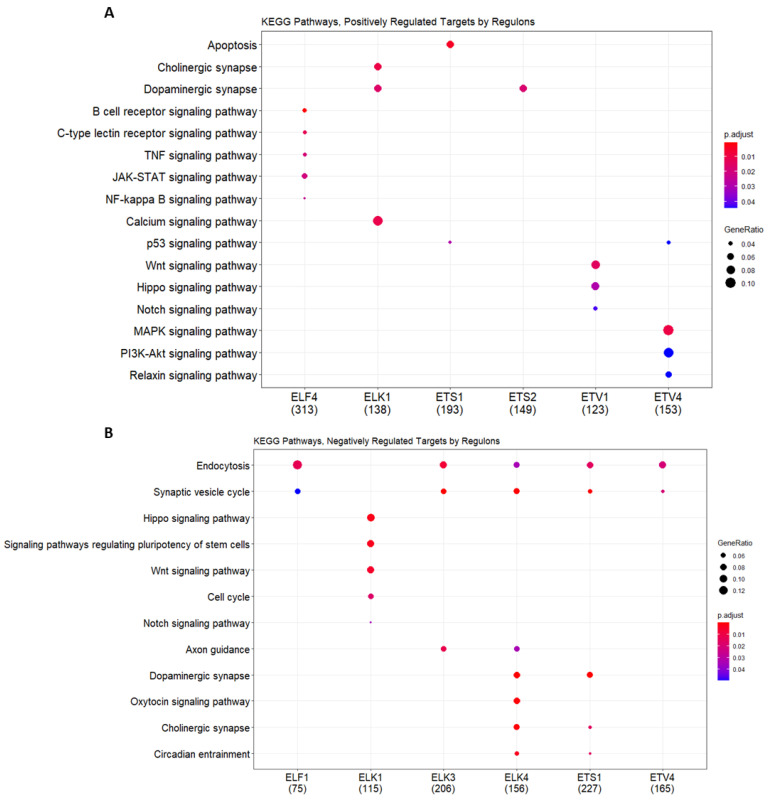
KEGG enrichment analysis of individual regulons and their (**A**). positively and (**B**). negatively regulated targets. KEGG analysis was performed by clusterProfiler with an adjusted *p*-value < 0.05. The gene ratio indicates the number of genes enriched with corresponding KEGG pathways among the total gene number introduced into the enrichment analysis.

## Data Availability

The data is available in Appendix A and upon request.

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
