# Peer review of "Gene Regulatory Network of ETS Domain Transcription Factors in Different Stages of Glioma"

_jpm, 2021, doi:10.3390/jpm11020138_

Round 1

Reviewer 1 Report

This manuscript by Kurnaz and coworkers reports a comparative microarray analysis of patient glioma samples at various grades (2 to 4) of disease. It is aimed at identifying novel biomarkers for grade identification. The patient data was also compared with previous microarray studies where several ETS members were over-expressed in neuroblastoma cell lines.

GENERAL COMMENTS

Gliomas are one of the most common types of primary brain tumors. Effective molecular approaches to grading is therefore a clinical priority. In my view, this is a significant objective. The data appears to show note-worthy differentiation of glioma samples from the non-tumor control, as well as a gradient from Grade 2 to 4. This much appears promising. Figure 1A is a strong piece of result. Looking beyond ETS proteins, what can the authors share in terms of the identities and functions/pathways of the differentially expressed genes?

My understanding of the authors’ writing is that the data from the SH-SY5Y cell line is used to help filter down the patient data and highlight strongly perturbed targets. However, would this have an effect on biasing the patient results towards ETS proteins that are specifically overexpressed (Elk1 and PEA3) in the cultured cell data? From Figure 2E and elsewhere, members in the Elk and ETV subfamilies are primarily represented, but other subfamily members such as PU.1 are not. Since PU.1 and several other ETS members that are tissue-specific and contribute to glial cell function, their absence from the authors’ attention is a major concern. I highly encourage the authors to address this gap. 

In terms of presentation, the bioinformatic results need much more in-depth analysis and explanation.  Several of the figures, appearing to contain much information, are discussed only superficially in the text e.g., Figures 4 and 5. These figures are nice to look at, but frankly I doubt the average reader would be able to discern anything without active help from the authors.

SPECIFIC COMMENTS

  1. Much more information is needed on the clinical samples in terms of general patient characteristics and how pathology was performed.
  2. Figure 5: More interpretation on Figure 5. “These results suggest that enriched ETS regulons have both unique and common gene targets in gliomas” Could the authors comment on the degree to which the ETS relatives are differentiated? What about as a function of glioma grade? All the panels look superficially very similar. Also, fix axes for ETS2.
  3. Figure 6: Similar complaint as Figure 5.  Very splashy, what specific insight does the reader take from the graph?
  4. Page 14: “ETS proteins are known to exhibit little tissue specificity, and in fact many family members are ubiquitously expressed” is not generally true.  Class III members are a good example, and they are conspicuously absent from the results, which need attention (vide supra).

Author Response

Reviewer 1

  1. The referee has inquired whether we could share genes beyond ETS superfamily members in this study. Indeed, Functional Enrichment Analysis of differentially expressed genes through the grades of glioma shown in Figure 3 represents the general biological entities and pathways associated with glioma progression beyond the limit of ETS proteins. We have emphasized some of these biological processes and pathways in the text related to Figure 3, such as TGF-b signaling pathway, c-Myc and p15 genes, as well as genes in Notch signaling pathway, to name a few (page 5, lines 206-213).
  2. The reviewer has stated that filtering down the patient data using information from SH-SY5Y-based microarray analysis could bias the patient results towards ETS proteins where Elk-1 and Pea3 are overexpressed. We have, in fact, initially analyzed the patient data as is, and focused on the ETS expression levels without any filtering. Thereafter, we have used the SH-SY5Y microarray results in order to narrow down our analysis of ETS regulatory networks in patient samples (Fig.1, SH-SY5Y microarray data used prior to GNR analysis only). This filtering is not biasing the patient results towards ETS proteins, but mainly narrowing down the entire and extensive gene regulatory network of ETS proteins of the patient samples. (We would also like to note in this context that another manuscript we have submitted to this special issue in fact validates the SH-SY5Y based microarray results across patient GBM samples, GBM cell lines, and neuroblastoma cell lines, and of the target genes validated we can assure the reviewer that although cell context-specific variations occur, the general profile of regulations by Elk-1 is similar across cell lines; Savasan-Sogut et al, under revision in JPM).
  3. The reviewer also points out that from Figure 2E and elsewhere, members in the Elk and ETV subfamilies are primarily represented, but other subfamily members such as PU.1 are not, and the reviewer also points out that since PU.1 and several other ETS members that are tissue-specific and contribute to glial cell function, their absence from the manuscript is a major concern. We have indeed focused on not the entire ETS superfamily, but a selected set of subfamilies that have been reported in the literature in relation to gliomas, such as ETS, TCF, ELF, PEA3 and TEL subfamilies, but not SPI, PDEF, ERF, ESE, ELG or ERG subfamilies where little or no report was found in relation to gliomas. (Many of members of these neglected subfamiles in fact were found to be non-significant in our analyses). The reviewer is correct that PU.1 of the SPI subfamily is missing, and there is indeed one report in the literature that we could find on the role of PU.1 in progression of glioma (Xu et al, 2018), but not of the other SPI subfamily members SPIB or SPIC. We have included PU.1 expression level to the revised Figure 2 in this context, but mentioned in the text that we have focused our attention to the previously mentioned subfamilies for the rest of the study.  Relevant references have now been included in the revised text (page 5, lines 182-193).
  4. The reviewer has also commented that the bioinformatic results and figures need to be better explained. We hope that the above-mentioned revisions and expanded text will now be satisfactory. We would like to thank the reviewer for this improvement.
  5. The reviewer also suggested that much more information is needed on the clinical samples in terms of general patient characteristics and how pathology was performed. The dataset analyzed in this manuscript was obtained from a publicly available database of Gene Expression Omnibus. The published the dataset used in this study, the authors described that the clinical samples were obtained from surgery patients of Henry Ford Hospitals. Although the main characteristics and pathology were not defined in the original paper, they suggest that the clinical samples were diagnosed for glioma grades by using WHO standards. This information is now included in the revised manuscript.
  6. The reviewer has requested more interpretation on Figure 5, and pointed out that the sentence “These results suggest that enriched ETS regulons have both unique and common gene targets in gliomas” was not clear. In the gene regulatory network inference algorithms performed in this study, Figure 5 represents the 2-tailed Gene Set Enrichment analysis (GSEA) of ETS regulons obtained from the initial transcriptional network that shown in Figure 4. The initial network obtained from the algorithm is very large and does not represent any statistical inferences. To obtain a more focused network including the differentially expressed genes, TNA algorithm and GSEA were performed. 2-tailed analysis was also used in order to determine negative and positive co-expression patterns of target genes with their regulons. In this figure, genes were ranked for its fold change shown in x axis, and other enrichment results were shown in y axis. Common and distinct targets were shown in the positive and negative classification in the graphs as a color code. In depth representation of this figure were shown in the next figure (Figure 6) as a network, the distinction of regulons targets clustered in the regulatory network as distributed all colored circles in the Figure 6. The function of these common and unique targets obtained with functional enrichment analysis were represented in Figure 7 and 8. We hope these explanations are satisfactory to the reviewer.  We have also corrected the ETS2 graph of Figure 5.
  7. The Reviewer also pointed out a similar issue with Figure 6, requesting a more detailed explanation. To clarify the investigation of common  and unique targets of ETS regulons resulting from gene regulatory network inference, networks are a useful source of visual representation. In the Figure 6, readers could investigate the distribution of differentially expressed genes regulated by ETS regulons with their fold changed with nodes color codes and interaction with ETS regulons by edge color codes in the network. However, to enhance the understanding of the gene regulatory network and increase the reproducibility the all information about the network were included into manuscript as a table in the supplementary material (Table S7).
  8. The reviewer points out the statement on page 14, “ETS proteins are known to exhibit little tissue specificity, and in fact many family members are ubiquitously expressed”, to be not generally true, for example Class III that is absent from the study. Indeed, we have focused on glioma-related subfamilies of class I (ETS, TCF, PEA3) and class II (ELF, TEL) in this study, which is now clearly stated in the revision.

Reviewer 2 Report

In their manuscript, "Gene regulatory network of ETS domain transcription factors in different stages of glioma," Babal et al. classified and identified the stage- or grade-dependent biomarkers from human glioma microarray samples using transcriptomic profiling and gene regulatory network inference algorithm. The author further compared the data with other microarray studies in neuroblastoma cell lines.

The data shows that ETS genes including ETV1, ELK3, ETV4, ELF4 and ETV6 can be used as novel biomarkers for identification of different glioma grades. 

The manuscript is well written, and the results support the conclusion.

There are some points that the authors should address.

  • Some abbreviations have not been defined for example CNS, Fli1, IDH.
  • The data obtained in this manuscript using epilepsy samples as control. If the authors can further discuss on the ETS gene expression among the different types of epilepsy, this will strengthen the quality of this manuscript.  
  • Figure 3 legend is in Italic style, please correct it.

Author Response

Reviewer 2

We would like to thank the reviewer for kind comments.  We have tried our best to make the minor revisions suggested by this reviewer as follows:

  1. Some missing abbreviations were included into manuscript as suggested.
  2. The italic style of Figure 3 legend was corrected.
  3. The dataset analyzed in this manuscript was obtained from a publicly available database of Gene Expression Omnibus. The published the dataset used in this study, the authors described that the clinical samples were obtained from surgery patients of Henry Ford Hospitals. Although the main characteristics and pathology were not defined in the original paper, they suggest that the clinical samples were diagnosed for glioma grades by using WHO standards. The main reason of using epilepsy samples as control could be the limitation of obtaining brain tissue from healthy people by using surgery. Due to limitation of transcriptomic study, we found that epilepsy dataset including different brain tissue. Additionally, the expression of ETS members were investigated in this dataset (Supplementary figure S1).
